# Hybrid Gold Nanorod-Based Nanoplatform with Chemo and Photothermal Activities for Bimodal Cancer Therapy

**DOI:** 10.3390/ijms232113109

**Published:** 2022-10-28

**Authors:** Lilia Arellano-Galindo, Eva Villar-Alvarez, Alejandro Varela, Valeria Figueroa, Javier Fernandez-Vega, Adriana Cambón, Gerardo Prieto, Silvia Barbosa, Pablo Taboada

**Affiliations:** 1Grupo de Física de Coloides y Polímeros, Departamento de Física de Partículas, Facultad de Física, Instituto de Materiales (IMATUS) e Instituto de Investigaciones Sanitarias (IDIS), Universidade de Santiago de Compostela, 15782 Santiago de Compostela, Spain; 2Nanostructured Funtional Group, Catalonian Institute of Nanotechnology (ICN2), Universidad Autónoma de Barcelona Campus, Av. Serragalliners s/n, 08193 Barcelona, Spain; 3Departamento de Ingeniería Química, CUCEI, Universidad de Guadalajara, Guadalajara 44100, Mexico; 4Grupo de Biofísica e Interfases, Departamento de Física Aplicada, Facultad de Física, Instituto de Materiales (IMATUS), Universidade de Santiago de Compostela, 15782 Santiago de Compostela, Spain

**Keywords:** gold nanorods, layer assembly, photothermal therapy, multimodal therapy, cell death pathway

## Abstract

Metal nanoparticles (NPs), particularly gold nanorods (AuNRs), appear as excellent platforms not only to transport and deliver bioactive cargoes but also to provide additional therapeutic responses for diseased cells and tissues and/or to complement the action of the carried molecules. In this manner, here, we optimized a previous developed metal-based nanoplatform composed of an AuNR core surrounded by a polymeric shell constructed by means of the layer-by-layer approach, and in which very large amounts of the antineoplasic drug doxorubicin (DOXO) in a single loading step and targeting capability thanks to an outer hyaluronic acid layer were incorporated by means of an optimized fabrication process (PSS/DOXO/PLL/HA-coated AuNRs). The platform retained its nanometer size with a negative surface charge and was colloidally stable in a range of physiological conditions, in which only in some of them some particle clustering was noted with no precipitation. In addition, the dual stimuli-responsiveness of the designed nanoplatform to both endogenous proteases and external applied light stimuli allows to perfectly manipulate the chemodrug release rates and profiles to achieve suitable pharmacodynamics. It was observed that the inherent active targeting abilities of the nanoplatfom allow the achievement of specific cell toxicity in tumoral cervical HeLa cells, whilst healthy ones such as 3T3-Balb fibroblast remain safe and alive in agreement with the detected levels of internalization in each cell line. In addition, the bimodal action of simultaneous chemo- and photothermal bioactivity provided by the platform largely enhances the therapeutic outcomes. Finally, it was observed that our PSS/DOXO/PLL/HA-coated AuNRs induced cell mortality mainly through apoptosis in HeLa cells even in the presence of NIR light irradiation, which agrees with the idea of the chemo-activity of DOXO predominating over the photothermal effect to induce cell death, favoring an apoptotic pathway over necrosis for cell death.

## 1. Introduction

Nowadays, the systemic administration of chemodrugs to kill malignant cancerous cells, so-called chemotherapy, is the preferred treatment of choice to fight against different types of tumors such as breast, lung, gastric, colon and ovarian at different stages of development [1] with relative success. Nevertheless, the high doses and prolonged duration of treatments are usually accompanied by severe adverse side effects, for example, toxicity in healthy tissues consequence of non-specificity, development of drug resistances, and/or achievement of suboptimal therapeutic concentrations in the diseased cells/tissues due to poor drug diffusion and accumulation conditioned by cancer structure and vascularization. In addition, the impossibility to early detect tumors of very reduced size with currently available bioimaging diagnostic techniques, or circulating malignant cells in blood through liquid biopsy accurately also involves a loss of therapeutic efficiency of current cancer therapies [2]. For this reason, there is an urgent need to develop new alternative therapies for cancer treatment.

To protect chemodrugs from the hostile physiological environment, favoring their localization and concentration in the tumoral diseased areas while precluding their recognition and clearance by the reticulo-endothelial system, nanocarriers of different natures such as liposomes, polymeric and inorganic nanoparticles (NPs), protein and viral capsids [3] appear as excellent candidates to solubilize, transport and delivery chemodrugs to their site of action thanks to their nanometer size and subsequent selective accumulation in the diseased areas through the enhanced permeation and retention (EPR) effect [4], possibility to evade opsonization and macrophage recognition leading to extended circulation times, to bear or being functionalized with additional moieties to bind overexpressed receptors in cancerous cells [5,6], and even to provide additional therapeutic modalities thanks to their composition and/or physical or biochemical properties.

In this regard, among the different types of nanocarriers able to simultaneously transport chemodrugs but also to provide additional therapeutic outcomes, gold nanoparticles (Au NPs) appear as an interesting alternative thanks to their controllable size and morphology, which allow the tuning of their outstanding optical properties, i.e., the generation of localized surface plasmons and subsequent ability to transform absorbed light of suitable wavelength into heat [7], which may allow their use as, for example, photothermal agents [8,9]. In addition, this type of inorganic NP is colloidally stable, nontoxic, nonreactive, and allows facile surface functionalization through well-established (bio)chemical methods, among other advantages [10]. Amongst the different types of Au NPs, Au nanorods (AuNRs) have been largely studied thanks to their anisotropic structure, which may favor their internalization within different types of cells [11], and their ability to transform absorbed light within the first biological window (near infrared region, NIR) into localized heat with high efficiency as a consequence of their large absorption cross section [12], thus reducing both the energy required for light stimulation for photothermal treatment (either photoablation when T > 45 °C, or tumor sensitization when T < 45 °C) and the affectation to surrounding healthy cells/tissue. Additionally, thanks to their large available surface areas, AuNRs are attractive nanocarriers for transport chemodrugs, photosensitizers, small biomolecules, and genetic material showing much larger available surface areas for such purposes than other nanocarriers [13]. In this regard, to use AuNRs as delivery platforms with success, several issues must be previously overcome, including the toxic nature of the positively charged CTAB surrounding layer resulting from the NP synthetic process; non-specific cargo delivery from the NP; unspecific targeting to healthy cells and tissues; and particle opsonization and subsequent body clearance. Hence, different strategies of AuNR surface functionalization have been developed in recent years to minimize/solve such drawbacks as ligand exchange [14], polymer adsorption [15], silica coating [16], etc. [17], which should serve to eliminate or at least hide/mask the underlying CTAB layer as well as provide direct targeting capabilities by means of the easy attachment/bioconjugation of targeting ligands as antibodies, peptides, proteins, sugars, aptamers, etc. [18,19]. In addition, the functionalizing layer should enable the controlled regulation of payload release either by internal (pH, temperature, presence of enzymes, osmolality, levels of oxygen, etc.) and/or external (light, electrical and magnetic fields, ultrasounds, etc.) stimuli without compromising the colloidal stability of the particles in the biological milieu [20]. 

Therefore, a multimodal therapeutic approach that simultaneously combines the chemotherapeutic effect of an antineoplasic drug encapsulated within a AuNR platform providing a strong plasmonic photothermal under suitable NIR light irradiation conditions appears an outstanding alternative to provide synergistic therapeutic benefits in the treatment of cancerous cells by providing different and complementary routes to induce cell toxicity as well as opening a new way to overcome drug resistances. In this sense, different previous studies have exploited such an approach [21,22,23]. In one of them, we previously proposed the use of AuNRs able to encapsulate the chemodrug doxorubicin (DOXO) or siRNA and combine its potential therapeutic effect with plasmonic photothermal therapy (PPTT) for efficient cancer treatment [24,25]. To achieve suitable cargo loading and the platform’s sufficient colloidal stability, the NPs surfaces were modified following a layer-by-layer (LbL) assembly technique using polymers of different ionic nature (negatively charged poly(styrene sulfonate), PSS, and positively charged poly(lysine), PLL, respectively), and the cargo molecules included in the assembled polymeric layer by exploitation of the electrostatic interactions between the underlying polymeric layer (either PSS for DOXO or PLL for siRNA, respectively). It was observed that relatively important concentrations of the drug could be incorporated in the nanoplatform, and its release was controlled either by the progressive enzymatic-triggerable disassembly of the coating polymeric layer in the presence of proteases or by the application of external NIR light irradiation under suitable illumination, which influence the strength of attractive interaction between the NP coating layers. The addition of a final outer external coating layer composed of hyaluronic acid (HA) surrounding the AuNRs involved a selective targeting to cancerous cells overexpressing the CD44 receptor, and the simultaneous combination of both chemo- and photothermal therapies led to a synergistic and targeted killing effect in cancerous cells under suitable illumination conditions. Nevertheless, other important factors include the optimization of the AuNR localized surface plasmon bands to match the incident NIR illumination wavelength and subsequent improvement of photothermal conversion activity; the achievement of larger amounts of encapsulated drug in a single step at lower fed initial concentrations; the regulation of cargo release under the simultaneous application of different triggers; or the efficacy and role played by each individual therapeutic approach when simultaneously applied in a multimodal therapeutic treatment and their influence on the induced cell death mechanisms was not considered. 

Hence, in this work, we synthesized AuNRs in high yields with tunable NIR localized plasmon surface resonance bands, LPSR bands, from ca. 700 to 816 nm for efficient PPTT, and particle lengths and widths within a very narrow range of 32–36 and 8–12 nm (AR variation from ca. 2.6 to 4.1), respectively. As expected, the plasmonic photothermal heat generation of the different AuNRs increased as their LPSR progressively matched the irradiated NIR light wavelength, but the loss of heat generation efficiency by off-resonant LPSRs followed an exponential decay, with only an exceptional 25% heat loss within an off-resonance range of 70 nm departing from the NIR irradiation wavelength. To provide cytocompatibility as well as colloidal stability, AuNRs of 802 nm were surface-functionalized using the LbL assembly technique by sequential deposition of alternating layers of negatively charged PSS and enzymatically, biodegradable and positively charged PLL, with a final outer surface layer composed of HA to provide with targeting capabilities to certain cancerous cells. Prior to PLL deposition, the chemodrug DOXO was incorporated under suitable solution conditions to maximize its loading, reaching up to ca. 350 μg/mL with an encapsulation efficiency (EE%) of 70% in a single drug coating step. DOXO release was observed to be negligible at early stages of incubation thanks to the compact packing of the polymeric surface coating, and when simultaneously triggered by enzymatic degradation of the PLL layer and external applied NIR light, it gives rise to a characteristic stair-like release profile, which confirms perfect control over the cargo diffusion out of the nanoplatform. Finally, the analysis of the multimodal therapeutic ability of the developed nanoplatform demonstrated that the combination of both chemo- and photothermal therapy provide a higher ability to kill malignant cells in a selective manner to high extents, apoptosis being the main cell death mechanism involved even when NIR light is largely applied. 

## 2. Results and Discussion

### 2.1. Synthesis and Photothermal Characterization of the Metallic Cores

AuNRs of different aspect ratios (ARs) were obtained within a very narrow size range by means of a one-step silver-assisted seeded growth methodology [26] using 2–3 nm cetyltrimethylammonium bromide (CTAB)-capped Au NPs as seeds (see Experimental Section and Appendix A, SM, for further details). To tune the AuNRs AR, only the amount of trace silver ions added to the Au growth solution was changed. A very fine-tuning of the AuNRs AR between 2.5 and 4.1 could be achieved (Figure 1a–d) through tiny changes between 32 and 36 nm in NP length and 8 and 12 nm in NP width. Appendix A shows the cumulative frequencies for length, width and AR of some different obtained AuNRs as measured by TEM. From DLS, hydrodynamic radii of AuNRs were found to be between 14.4 ± 0.5 nm and 17.1 ± 0.4 nm. It is worth recalling that light-scattering analysis (using the Stokes–Einstein relationship, see Materials and Methods for details) assumes that NPs are spherical; thus, the current sizes determined by DLS cannot be taken as real but only as an average of the NP length and width in all possible particle orientations in solution [27,28]. 

Reaction NP yields (i.e., the number of particles with AR > 1.9 divided by the total number of particles measured) of 67%, 64%, 70%, 53%, 89%, 90%, 95%, 96% were obtained for AuNRs with longitudinal LSPRs centered at 700, 717, 735, 755, 766, 785, 792, 802 and 816 nm, respectively (see Figure 1e). The progressive increase in silver concentration involved a more anisotropic structure, in agreement with previous reports [29,30,31,32,33] by hardly affecting the AuNRs length but decreasing their width, thus involving a linear LSPR redshift of up to ca. 120 nm as the AuNR AR changed from 2.6 to 4.1 (Figure 1f,g). Burrows et al. [34] previously showed that the AgNO_3_ concentration would seem only to have a certain influence on AuNR lengths but not on their widths. Here, it is observed that the AuNRs length hardly changes, but the width did slightly (Appendix A), which may originate from the CTAB micelles surrounding AuNRs, whose size and shape are affected by different factors as concentration, presence of salts, temperature, etc. Here, we used a larger surfactant concentration and temperature than in previous studies, which can be decisive for the final NP dimensions [35].

On the other hand, the observed temperature increases under NIR irradiation were concentration- and fluency-dependent with shifts ranging, for example, from ca. 0.8 to 5.6 °C and 3.2 to 16.0 °C for particles under 0.25 and 3.0 W/cm^2^ at a concentration of 1·10^8^ and 1·10^11^ NPs/mL, respectively, for AuNRs with longitudinal LPSR centered at 802 nm (Appendix A), then allowing the definition of the required conditions to photothermal-induced cell death expected by apoptosis or necrosis pathways (Figure 1i). It was also observed that the longer the departure of AuNR LPSRs from the NIR excitation wavelength (808 nm), the lower the temperature increase was obtained (Figure 1h). This was confirmed for AuNRs with LPSR maxima below 760 nm; however, the 766 nm ones provided a rather similar final macroscopic temperature than 816 nm ones. The loss of heat generation efficiency by off-resonant AuNRs was quantified by correlating the temperature increments (Δ*T*) with the off-resonant LPSR shifts, Δλ = │λ_laser_ − λ_LSPR_│. An exponential decay of Δ*T* as the LSPR band blueshifts from the light wavelength could be observed (Figure 1j), but surprisingly, a temperature difference as small as 3.0 °C was noted when the AuNRs LPSR maximum blue-shift 70 nm away, thus enabling an excellent photothermal activity of out-of-resonance AuNRs within this range. To achieve similar Δ*T* using AuNRs with smaller LSPRs maxima (Δλ > 70 nm), large increases in NP concentration or laser fluency would be required (see Appendix A), which can preclude their use in photothermal therapeutic applications [10,36]. 

To put more light in photothermal-induced heating provided by AuNRs, we developed a computer simulation of the heat distribution profiles upon excitation of one AuNR with longitudinal LPSR band centered at 802 nm and dimensions 32 × 8 nm under 808-NIR light excitation of 1.0 W/cm^2^. For the simulation, initially, the AuNR was irradiated with light of different polarization, and its absorption efficacy and corresponding energy emission spectra were recorded (Appendix A). Both absorption and emission show maxima at the NIR wavelength length when the light polarization coincides with the longitudinal direction of the NP, and progressively decreases as the polarization changes. This is also reflected in the energy released heatmap, where a high amount of energy is released to the surrounding AuNR environment when this is longitudinally illuminated along its major axis in contrast to transversal irradiation. In addition, in the former case, the energy is released along the whole particle symmetry but is slightly more concentrated in the middle of the NP, whereas in the latter case, the energy diffusing out the particle is only observed at the center of the AuNR but not in its edges (Figure 2a,b). This can be additionally confirmed by the heat diffusion maps obtained at 5 and 25 nm away the NP. 

Figure 2c shows that the temperature enhancement in the AuNR surroundings can reach up to 60 °C when longitudinally irradiated at a particle distance of 5 nm, and the heatmap displays an elongated profile similar to that of the particle; moreover, the temperature increase is more uniform at the particle edges than in its center, probably as a result of the energy crystal lattice vibrations from the latter to the former ones, where there is a larger lattice strain [37], leading to a slightly larger heat release. Conversely, when transversally irradiated, the heatmap is quasi-spherical and with maximum temperature increases of only up to ca. 0.03 °C (Figure 2d). Similar observations are also obtained when increasing the distance to the NP to 25 nm, but lower temperatures are reached (Figure 2e,f).

### 2.2. Functionalization, Colloidal Stability and Photothermal Conversion of the Metallic Platform

For an optimal photothermal conversion while using particle concentrations and NIR fluencies within tolerable and cytocompatible limits, for further work, we used the obtained AuNRs with LPSR centered at 802 nm and dimensions of 32 × 8 nm due to the expected slight redshift of the LPSR band (see below) upon NP functionalization to well match the CW laser wavelength used for irradiation (808 nm), while keeping a high absorption cross section. Particle functionalization was made by means of the well-established LbL coating technique using the following deposition sequence: anionic PSS, cationic DOXO drug, cationic PLL and anionic HA by exploiting the electrostatic interactions between components to obtain either PSS/PLL/HA- or PSS/DOXO/PLL/HA-coated AuNRs when corresponding. To achieve an optimal stability and size as well as to enhanced and optimize DOXO loading, the polyelectrolyte (PE) layer deposition was optimized and made in the presence of 12 mM NaCl in an acidic solution of pH 4.1 containing KOH (see Appendix A and additional information in SM). The successful coating of the different layers was corroborated by slight redshifts observed in the LPSR band (Figure 3a,b) and the changes in particle ζ-potentials from positive to negative and vice versa upon subsequent polymeric layer deposition (Figure 3c). FTIR spectra additionally confirmed the successful polymeric coating onto the NPs (Appendix A). The sequential deposition of the different polymeric and drug layers led to slight increases in the platform sizes but without signs of aggregation (Figure 3d–f). The selected incubation condition also allowed a very efficient incorporation of DOXO to high extents, and drug coating/encapsulation onto the particles remain stable at the selected conditions (ca. 70% up to 350 μg; see also Appendix A and further details in SM). On the other hand, the AuNR-based nanoplatforms were colloidally stable after long incubation at basic and physiological pH (9.0 and 7.4, respectively) but became relatively more unstable under acidic conditions, as denoted by the progressive increases in platform size (see Figure 3g) and variations of ζ potential values (Appendix A) within the first 48 h of incubation due to the reduced ionization of the negatively charged carboxylic groups of PSS and HA, which diminishes the attractive electrostatic attractions with positively charged PLL. The presence of serum proteins also involved certain size increases of the nanoplatform as a consequence of protein corona formation within the first stages of the incubation process, which may also lead to some platform clustering as observed from hydrodynamic sizes (Figure 3h and Appendix A). The inclusion of a protease as trypsin should progressively degrade the PLL layer, allowing an endogenous triggered cargo delivery mechanism (see below); however, for this release mechanism to be truly effective, the stability of the platform must be maintained. Figure 3i and Appendix A showed than when the protease is added to the solution medium, no subtle changes in the platform hydrodynamic size and smaller variations in ζ-potentials compared to free-trypsin condition are observed within the first days of incubation, in agreement with the protective role played by the HA outer layer, which impedes a rapid contact of the protease with the PLL coating. Such behavior is kept for 96 h, in which a stabilization of the NP surfaces took place by additional adsorption of proteins from the supplemented serum. Conversely, at longer incubation times (>4 days), the biodegradation of the PLL biopolymeric layer might be very important, giving rise to a certain destabilization of the coating layer, leading to the formation of particle clusters, as observed in the changes in the nanoplatform sizes. 

Next, the photothermal conversion efficiency of the nanoplatform was assessed. To do that, three consecutive heating/cooling cycles were performed with a solution of 1.0·10^11^ NPs/mL AuNRs at different fluencies (1.0, 2.0 and 3.0 W/cm^2^) for several minutes (see Figure 3j). It is observed that after the successive irradiation cycles, the platform does not change their transverse and longitudinal plasmon bands and maxima (Appendix A), which correlates with the absence of large morphological changes (Appendix A). The heat profiles display a profile composed of a pronounced temperature increase of the solution for the first minutes of irradiation, followed by a quasi-plateau phase when the generated heat by the light irradiation and that dissipated in the particles’ surroundings reach equilibrium. At the highest fluency tested (3.0 W/cm^2^), it is observed that the maximum temperature achieved during the third cycle is slightly lower than the other, probably due to some particle reshaping and/or destabilization of the polymeric coating [38] (Appendix A); in addition, the heating profile shows a shoulder at short incubation times, probably mediated by an initial heat release saturating the solution, which promotes certain evaporation inside the test tube. On the other hand, after the laser light is off, the solution temperature exponentially decreases until reaching the initial values. The photothermal conversion efficiency (η) and the characteristic heat velocity rate (τ_s_) after each irradiation cycle were determined using the heat transfer model developed by Roper et al. [7]. It is worth mentioning that η is an intrinsic property of the nanoplatform; that is, it depends on particle size, morphology and composition, whereas τ_s_ depends on the experimental configuration, that is, on the cell dimensions, solution volume, etc. [39,40]. The obtained η values laid in the range 0.41–0.45, 0.60–0.71 and 0.40–0.55 at 1.0, 2.0 and 3.0 W/cm^2^ for the different irradiation cycles, and are in agreement with previously reported data. For example, Pattani et al. [9,41] and Cole et al. [39,40] reported photothermal conversion efficiencies of ca. 0.50 and 0.55 for AuNRs with dimensions 26 × 7 nm and 44 × 13 nm and LPSR maxima located at 770 and 780 nm, respectively (see Table 1). η values progressively decrease with the number of applied light irradiation cycles at a fluency of 3.0 W/cm^2^, which agrees with the observed heating profile and certain melting of the edges of the metallic particle (Figure 3j and Appendix A).

### 2.3. DOXO Release Profiles

DOXO release kinetics from PSS/DOXO/PLL/HA-coated AuNRs in physiological mimicking media with supplemented 50% (*v*/*v*) FBS at pH 7.4 and pH 5.0 in the presence of simultaneous internal enzymatic and external light triggers showed stair-like profiles, particularly at incubation times lower than 15 h (Figure 3k,l). Such behavior stems from the development of small bursts after the application of the NIR light stimuli of 1.0 and 2.0 W/cm^2^ for 5 min at 2, 5, 9, 12 and 24 h of incubation. At longer incubation times, the release profiles progressively levelled off to provide a more sustained release, since most of the drug was already diffused out of the nanoplatform as a consequence of the disturbance of the polymeric coating by both thermal and enzymatic degradation. It can be observed that the combination of both the protease and light-assisted release led to a faster release kinetics when compared to the individual applied stimuli, with up to ca. 83 and 71% and 100% and 96% of encapsulated DOXO released under 1.0 and 2.0 W/cm^2^ under NIR light irradiation at pH 7.4 and 5.0, respectively, after 96 h. These values contrast with ca. 46% and 53%, and 55% and 60% obtained in the presence of either trypsin or applied NIR light (1.0 W/cm^2^) at both pHs, respectively. Moreover, the presence of the light external stimulus makes the initial lag phase observed in release profiles at short incubation times disappear once applied, allowing a tight control over the released drug concentration at initial steps (up to ca. 12 h). Unfortunately, such outstanding control cannot be completely kept at longer times due to the progressive particle coating destabilization produced with additional irradiations. Light irradiation serves to generate localized heat which helps in largely modifying PE-drug interactions, thus, facilitating DOXO release [42]. Additionally, such modification is much more effective when it is combined with the activity of a protease as trypsin, which attacks peptide bonds and, thus, progressively degrades the PLL layer (see Figure 3k,l). It is also worth mentioning that DOXO release is larger at pH 7.4 than at pH 5.0 in the presence of both the individual or combined stimuli, which can be related to the different conformation adopted by PLL at the different pH [43], influencing the electrostatic interactions and entanglements with the underlying PSS layer [25].

### 2.4. In Vitro Biological Evaluation

Although CTAB-coated AuNRs were toxic to cells at any concentration range, PSS/PLL/HA-coated ones possessed a concentration-dependent cell viability in cancerous cervical HeLa and non-cancerous 3T3-Balb fibroblasts cells, with cell survivals well above 60% and 75% at concentrations below 5·10^10^ NPs/mL after 24 and 48 h of incubation as demonstrated by the CCK-8 cytotoxic assay (see Appendix A). These data reinforce the biocompatibility of the nanoplatform. Hence, the polymeric coating strategy provides stability and stealthiness to the particles whilst masking and preventing the desorption of potential remaining CTAB onto the NP surface.

Next, PSS/DOXO/PLL/HA-coated AuNRs were tested as a potential platform for combined bimodal chemo- and photothermal treatment, with the aim of deciphering the contribution to therapeutic activity and induced mechanism of cell death related to each treatment as well as the synergistic or additive character of the bimodal therapeutic approach. To do that, additional CCK-8 cytotoxicity assays were performed at two incubation times (24 and 48 h) and different temperatures (4, 25 and 37 °C) to decouple the effects of the chemoactivity of DOXO and the macroscopic photothermal effect provided by the NIR light irradiation of the nanoplatform (5 min at 1.0 or 2.0 W/cm^2^) [44]. It is also worth mentioning that HeLa cells largely overexpress CD44 receptors on their surface able to specifically interact with great affinity with HA [45,46,47] in contrast to mouse fibroblasts, which do not express this type of receptor, thus enabling us to highlight the role of active targeting on the induced therapeutic activity. In these experiments, DOXO was incorporated into the nanoplatform at two different concentrations (75 and 100 µg), and free DOXO was used as positive control. In addition, PSS/PLL/HA- and PSS/DOXO-coated AuNRs (that is, particles with directly exposed drug to the physiological medium without further coating protective layers) were used as additional controls for photothermal activity and drug delivery modulation, respectively.

Figure 4 shows the toxicity of free DOXO, bare PSS/PLL/HA-, and cytostatic PSS/DOXO- and PSS/DOXO/PLL/HA-coated AuNRs loaded at two different drug concentrations in the absence and presence of NIR light irradiation (1.0 and 2.0 W/cm^2^ for 5 min) in HeLa and 3T3 Balb fibroblasts cells. It was observed that free DOXO-induced cell toxicity is larger in HeLa cells than in 3T3 Balb ones due to the larger metabolic activity of the former. Cell mortality also increases for HeLa cells as the incubation time does at ca. 12–18% after 48 h in the absence of irradiation at 37 °C (Appendix A). PSS/PLL/HA-coated AuNRs were non-toxic for both type of cells (viability > 90%) except for HeLa cells in the presence of a NIR fluency of 2.0 W/cm^2^, for which reductions in cell viabilities up to ca. 81% and 70% at 24 and 48 h of incubation were observed, respectively, due to the generated plasmonic photothermal effect. These decreases are less important than in previous studies, probably as a result of the relatively low particle NP concentration used in the experiments (1.0·10^10^ NP/mL) [25,48], and hence only at the highest irradiation condition and tumoral cell type could a slight temperature-dependent photothermal toxic effect be observed. 

On the other hand, when DOXO is incorporated in the nanoplatform at two different concentrations, PSS/DOXO/PLL/HA-coated AuNRs at 75 and 100 µg the combined effect of chemo- and photothermal activities leads to important reductions in HeLa cells survival compared to single light or chemotherapeutic applied treatments. Such mortality is observed to be both drug-concentration- and fluency-dependent. The incubation temperature also plays a key role to achieve the important observed cytotoxic effect under the application of the bimodal therapeutic approach; that is, at 4 °C cell cytotoxicity is within ca. 15–22% at both DOXO encapsulated concentrations and fluencies at 24 h of incubation, and progressively increases for HeLa cells as the temperature does from 4 to 37 °C reaching up to ca. 31 and 48% at 1.0 W/cm^2^ and 71 and 84% at 2.0 W/cm^2^ at the largest DOXO concentration at 37 °C after 24 and 48 h of incubation, respectively (Figure 4a–d). Hence, as the incubation temperature increases, and specially at 37 °C, the simultaneous application of the chemo- and photothermal therapies led to a synergistic toxic effect (a therapeutic outcome much larger than the sum of therapies applied individually), which benefits, on one hand, from the larger sensitivity of cells to the therapeutic activity of DOXO, the enhanced cell/tissue permeability and drug diffusion at larger temperatures [49] but also from the hyperthermia derived from the photo-stimulation of the nanoplatform, thus involving either cells’ photosensitization or thermal ablation depending on the reached macroscopic solution temperature. In this regard, departing from an incubation temperature of 25 °C, the maximum temperatures reached were 31 and 43 °C, whilst for departing from 37 °C, maximum temperatures were 39 and 46 °C under NIR fluencies of 1.0 and 2.0 W/cm^2^, respectively. Additionally, the combination of the larger encapsulated DOXO concentration within the nanoplatform and the largest fluency applied led to the largest cell cytotoxicities. In addition, the present data also corroborate that at rather low incubation temperatures, the photothermal effect does not provide a relevant additional cell toxicity added to the chemo-activity of the drug as observed from the data taken at 4 °C, confirming that under continuous illumination, the macroscopic temperature increase is the main responsible factor for inducing additional cell mortality rather than the nano/microscale ones, which, in this case, provide temperature increases below the therapeutic active threshold.

Finally, it is worth mentioning that the presence of the PLL/HA-coating layer also plays a key role in the therapeutic response. As observed from Figure 4, the absence of such biopolymer coating the DOXO layer involves a larger toxic effect of ca. 60% at 37 °C in the presence of 1.0 of NIR light irradiation, which is generally larger when compared to that of PSS/DOXO/PLL/HA-coated Au NRs. This behavior may confirm that the PLL/HA coating, apart from being a protective and targeting layer, also serves to diminish the rate of diffusion of DOXO out of the platform. This is additionally confirmed when comparing cell viabilities at 4 °C and 37 °C for which cell toxicities in HeLa cells are truly different (see for example, Figure 4a,b), since the coating layer is deeply altered at the largest temperature favoring drug release. That is, DOXO would need more time to achieve their complete diffusion out of the particles when the macroscopic temperature is lower, since more time is needed to achieve the requested energy to destabilize the interactions of the polymeric chains in the NP coating later [50].

Conversely, in 3T3 Balb cells, the administration of PSS/DOXO/PLL/HA-coated or PSS/DOXO AuNRs did not lead to relevant toxic effects after 24 h of incubation in the presence or absence of NIR illumination (Figure 4e–h and Appendix A); only at 48 h were cytotoxicities in the range of 20–25% observed, probably as a consequence of some drug diffusion through the cell membrane but with reduced cellular uptake and internalization of the whole nanoplatform, as also confirmed from ICP-MS data (Appendix A). This fact is confirmed from both the relatively scarce presence of NPs inside Balb cells observed by TEM (Figure 5a,b) and the rather low DOXO fluorescence in the cell cytoplasms and nuclei from fluorescence microscopy upon particle administration and incubation, even in the presence of NIR light irradiation (Appendix A); in fact, only after 12 h of incubation under irradiation of 2.0 W/cm^2^ can some DOXO be localized inside the nuclei. Moreover, some Au NRs can be discerned out of the cells without being internalized. Such behavior can be a consequence of the lack of CD44 receptors on the membrane surfaces of this type of cells, which precludes cellular uptake to high extents. In addition, it can be observed that Balb cells treated with PSS/DOXO/PLL/HA-coated AuNRs showed an unaltered morphology, whereas those subjected to free administered DOXO presented a more spherical shape denoting cell death, additionally confirming the lack of particle internalization (Appendix A) [51].

These observations are in contrast with the important NP uptake and internalization extent observed in HeLa cells as observed in fluorescence microscopy and TEM images. The TEM image in Figure 5c shows that the particles are inside the HeLa cell cytoplasm after 6 h of incubation, and under light irradiation, some cell dehydration and apoptotic bodies and blebs inside and around the cell membrane can be also discerned (Figure 5e), corresponding to both the chemoactivity of DOXO and plasmonic photothermal effect, in agreement with the Annexin-Propidium iodide assay (see below). A part of the observed particle-containing vesicles is shown as an enlarged image in Figure 5d, where some AuNRs can experience some melting inside after the application of the NIR illumination (Figure 5f). In addition, fluorescence microscopy images show that DOXO released from PSS/DOXO/PLL/HA-coated AuNRs appeared inside the cell cytoplasm and nuclei after 24 h of incubation, in agreement with a potential endocytic pathway of internalization for the platforms with a prolonged and sustained drug release (Figure 6b). This is contrary to the distribution of administered free DOXO earlier incorporated into the cell by diffusion and migrating into the nuclei, which agrees with the observed toxicity data (Appendix A). Moreover, it can be observed that DOXO release is progressively spread inside the cell with incubation time in the presence of NIR light irradiation, the DOXO release being particularly accelerated at the highest fluency, probably as consequence of a larger disruption of the nanoplatform coating layer (Appendix A). This fact further corroborates the control of the release kinetics by light.

Finally, the mechanisms involved in cell death by the application of a bimodal chemo- and photothermal treatment combining the antineoplasic DOXO and photothermal therapy were decoupled and elucidated by means of the Annexin V-propidium iodide assay by flow cytometry after 48 h of incubation at different incubation temperatures (4, 25 and 37 °C) and NIR irradiation conditions (1.0 and 2.0 W/cm^2^, 808 nm for 5 min; see Figure 7. Bare HeLa, PSS/PLL/HA-coated AuNRs and free DOXO were used as negative and positive controls, respectively. Healthy 3T3 Balb cells were also used as an additional control (see Appendix A).

The administration of free DOXO led to cell death by apoptosis, in the range of 70–90% and 60–80% for 3T3 Balb and HeLa cell, respectively, depending on the incubation conditions. For both types of cells, cell apoptosis slightly increases with laser fluency and incubation temperature as a result of certain temperature-dependent sensitivity of the chemo-activity of DOXO related to an enhanced diffusion of the drug at larger temperatures and enhanced chemoactivity, as commented above (see Figure 7). Moreover, the proportion of live cells is much smaller in HeLa cells as result of their accelerated metabolism in agreement with toxicity data; in particular, very large survival rates of 3T3 Balb at 4 °C are observed as the diffusion of the drug inside the cells slows down as consequence of reduced molecule mobility in solution. Meanwhile, for PSS/PLL/HA-coated AuNRs HeLa, cell death is mainly induced by necrosis at incubation temperatures of 25 and 37 °C, whereas at 4 °C, apoptosis is dominant (Figure 7a), the former effect probably a consequence of the disruption of cell membranes upon the temperature increase produced by the photothermal treatment. In this case, irradiation plays a key role in increasing the macroscopic temperature close to the ablation regime (>44–45 °C), particularly at 37 °C. Conversely, 3T3 Balb cells remains almost fully alive at similar bulk temperatures and irradiation conditions (Figure 7b), in agreement with the low internalization NP extents in this type of cells, which additionally confirms the importance of active targeting to achieve in this case an optimal cytotoxic activity with these nanoplatforms. 

Strikingly, PSS/DOXO/PLL/HA-coated AuNRs induced cell mortality mainly through apoptosis in HeLa cells even in the presence of NIR light irradiation in opposition to PSS/PLL/HA-coated ones (Figure 7). This would indicate that the chemo-activity of DOXO predominates over the photothermal effect to induce cell death, favoring and reinforcing the apoptotic pathway over the necrotic one. In fact, at incubation temperatures of 4 and 25 °C, the proportion of apoptotic cells is higher at 2.0 W/cm^2^ than at 1.0 W/cm^2^, in agreement with the sensitizing effect that the largest irradiation would exert at relatively low macroscopic temperatures, that is, with the macroscopic solution temperature being below the ablation regime (<45 °C), then favoring the pharmacological action of DOXO through its intercalation in the cellular DNA, blocking the action of topoisomerase II and avoiding the process of DNA replication, which causes tumor cells to enter in an apoptotic process. However, at 37 °C, the increase in macroscopic temperature by the plasmonic effect would explain the certain observed increase in necrosis regarding apoptosis, particularly at the fluency of 2.0 W/cm^2^. 

Conversely, in 3T3 Balb cells, the levels of necrosis and apoptosis are below 10–15% on average, in agreement with cytotoxicity data, and are independent of the incubation temperature or level of irradiation, except at 37 °C and 2.0 W/cm^2^ for which the levels of necrosis slightly increased (Appendix A). This behavior would be in accordance with the deficient internalization of the PSS/DOXO/PLL/HA-coated AuNRs observed in this cell line.

## 3. Materials and Methods

### 3.1. Materials

Hexadecyltrimethyl ammonium bromide for molecular biology (CTAB), tetrachloroauric acid (HAuCl_4_·3H_2_O), silver nitrate (AgNO_3_), sodium borohydride (NaBH_4_), poly(sodium-4-styrenesulfonate) (PSS) of molecular weight (Mw) ~70,000 g/mol, poly-L-lysine hydrobromide (PLL) of Mw ~22,000 g/mol, hyaluronic acid of M_w_ ~15,000 g/mol, and doxorubicin hydrochloride (DOXO) were purchased from Sigma-Aldrich. Ascorbic acid was from Fluka. Heat- inactivated fetal bovine serum (FBS), trypsin-EDTA (0.25X) and PBS pH 7.4 (10×) were from Hyclone (Thermo Scientific, Waltham, MA, USA). Prolong Antifade reagent with DAPI was from Molecular Probes, and the Cell Death Annexin V-Propidium Iodide kit was from Gerbu Biotechnik (Heidelberg, Germany). All other reagents were of analytical grade and/or suitable for cell culture as corresponding. All chemicals were used as received. Milli-Q water (Millipore, Burlington, MA, USA) was used throughout all the experiments.

### 3.2. Synthesis of AuNRs

AuNRs were synthesized using a seed-mediated growth method. First, CTAB-capped Au seeds were obtained. To do that, 7.5 mL of a 0.2 M CTAB solution was gently mixed with 0.25 mL of 0.01 M HAuCl_4_ in a water bath at 28 °C. As an indication, for both seeds and AuNRs preparation, CTAB was left at a constant temperature of 28 °C for one day under constant stirring of 200 rpm to achieve full solubilization and avoid foaming prior to use. Next, a 0.01 M NaBH_4_ solution was prepared in ice-cold water. This solution was left to rest for 2–3 min to ensure a good dispersion of the reductant. Afterwards, while the Au-CTAB solution was stirred at 200 rpm, 0.6 mL of ice-cold 0.01 M NaBH_4_ was added in one pull to the former, after which the mixed solution turned brownish yellow. This solution was mixed gently by hand for 2 min and then left undisturbed in a water bath at 28 °C for 1 h to allow the excess sodium borohydride to be decomposed. 

For the growth of AuNRs, 850 µL of a 0.01 M HAuCl_4_ solution was added to 20 mL of 0.2 M CTAB in a water bath at 28 °C, after which the solution turned bright yellow while stirred at 500 rpm. Then, 0.01 M of an AgNO_3_ solution was prepared in darkness. Different volumes of the silver solution (ranging from 126 µL to 246 µL) were added to the Au growth solution, followed by gentle mixing at 500 rpm. Then, 136 µL of a 0.1 M ascorbic acid (AA) solution was added, followed by gentle stirring at 500 rpm until the solution turned colorless (ca. 4 min). Finally, 220 µL of the Au seed solution was gently added to the Au growth solution while stirring for 2 min, and then stopped. The resulting solution was left undisturbed in a water bath overnight at 28 °C for ca. 12 h, becoming reddish-pink. The formed AuNRs were centrifuged at least twice at 28 °C for 30 min at ca. 14,500× *g* rpm and redispersed in 20 mL of deionized water. The UV-visible absorption spectra of the obtained AuNRs were measured using a Cary Bio 100 UV-vis spectrophotometer (Agilent Technologies, Santa Clara, CA, USA). The sizes and ARs of the AuNRs were measured using a JEOL JEM 1011 (Japan) transmission electron microscope operating at an accelerating voltage of 120 kV.

The AuNRs concentration per mL was calculated by means of inductively coupled plasma mass spectroscopy-MS (ICP-MS). For example, for the case of AuNRs with a longitudinal LSPR band centered at 802 nm, 1 mL of three different particle batches at an OD ~1 has an Au concentration of ca. 49.22 ± 0.37 µg/mL, taking into account the particle length and width obtained by TEM. The final concentration obtained under these conditions was of ca. 5·10^14^ rods/L. 

### 3.3. LbL Polymeric Coating of AuNRs

Multilayers of PSS, PLL and HA were successfully deposited onto the AuNR surfaces using the LbL methodology. For the PSS coating, a solution of PSS (10 mg/mL) in 12 mM NaCl (100 mL) was prepared. When this polyelectrolyte (PE) was completely dissolved, 1 mL of this solution was mixed with 1 mL of a 12 mM NaCl solution, and the resulting mixture was stirred at 500 rpm. Then, 1 mL of a AuNR solution (OD ~ 1) was added dropwise in the PSS solution while stirring. After 1 h of adsorption, the mixture was centrifuged twice at 15,000× *g* rpm for 20 min and resuspended in 1 mL of MilliQ water. Next, a solution of DOXO (1 mM) was prepared in an acetic acid buffer with KOH at pH 4.1 [52]. A suitable mass of DOXO was dissolved in 700 µL of the buffer solution and stirred at 500 rpm for 60 min. Next, 1 mL of the PSS-coated AuNRs was added dropwise to the DOXO solution under stirring. After 1 h, the mixture was centrifuged once at 13,000× *g* rpm for 10 min and the precipitate redispersed in 1 mL of MilliQ water. As the PSS/DOXO-coated AuNR particles have a negative surface charge, a subsequent cationic PLL layer can be used to coat the hybrids. Thus, 100 µL of a PLL solution (5 mg/mL) was added to 1 mL of MilliQ water and stirred for 5 min. Then, 1 mL of PSS/DOXO-coated AuNRs was added dropwise. After 1 h, the NPs were centrifuged at 15,000× *g* rpm for 15 min and resuspended in 1 mL of MilliQ water to the former solution. The deposition of the final HA layer was made by preparing 60 µL of a HA concentrated solution (1 mg/mL), which was mixed with 1 mL of MilliQ water and stirred at 500 rpm for 5 min. An amount of 1 mL of the former PSS/DOXO/PLL-coated AuNRs was added dropwise to the HA solution. After 1 h, the final coated AuNRs were centrifuged at 10,000× *g* rpm for 10 min and redispersed in 1 mL of MilliQ water.

### 3.4. Nanoplatform Characterization

#### 3.4.1. Dynamic Light Scattering (DLS)

DLS measurements were performed using an ALV-5000 digital correlator system (ALV 5000/E, ALV GmbH, Langen, Germany) equipped with a temperature control set at 25 °C. The scattered light was vertically polarized with a 532 nm solid-state laser (2 W). The hydrodynamic radius, R_h_, was obtained for diluted samples from DLS measurements at an incidence angle of 90° by analysis of the DLS data by means of the CONTIN algorithm developed by Provencher [53] and the Stokes–Einstein equation, R_h_ = k_B_T/6πηD, where k_B_ is the Boltzmann constant, T the temperature, η the solution viscosity, and D the diffusion coefficient of the particles. Measurements were performed at least in triplicate with a sampling time of 60 s each and averaged.

#### 3.4.2. Inductively Coupled Plasma-Mass Spectrometry (ICP-MS)

The Au concentration in solution or in cells was determined by inductively coupling plasma mass spectrometry (ICP-MS) in a Varian 820-MS equipment (Agilent Technologies, USA). 1 mL of 1·10^11^ AuNRs/mL (or 100,000 cells with NPs where corresponding) was dissolved in 0.3 mL HCl (37% (*v*/*v*)) and 0.1 mL HNO_3_ (70% (*v*/*v*)). Solutions were diluted with deionized water until reaching a final volume of 2 mL. The intensity of the emission wavelength was measured and compared to a standard solution.

#### 3.4.3. Electrophoretic Mobilities

ζ-potentials of bare and PE-coated AuNRs were measured using a Nano ZS (Nanoseries, Malvern Instruments, Malvern, UK) equipped with a 633 nm He-Ne laser of 4 mW power. The instrument measures the electrophoretic mobility of the particles and converts it into ζ-potential data using the classical Smoluchowski equation:α = ε ζ/η(1)
where α, ε, ζ, and η denote the electrophoretic mobility, permittivity of the medium, ζ -potential of the particles, and viscosity, respectively. Each sample was fed into a folded capillary, clear, disposable cell. Measurements were initiated after attaining thermal equilibrium at 25 °C. The number of runs for each experimental point was automatically determined by the software, and measurements were performed in triplicate. The results were reported as the mean ± standard deviation (SD). Solution with concentrations lower than 100 µg·ml^−1^ were used.

#### 3.4.4. Transmission Electron Microscopy (TEM)

To acquire transmission electron microscopy (TEM) images of Au NRs, a drop of 5 µL of nanoplatform sample was deposited onto carbon-coated copper grids, blotted, air dried, and then examined with a JEOL JEM 1011 (Japan) transmission electron microscope operating at an accelerating voltage of 200 kV. Samples were diluted when needed before deposition onto the copper grids.

#### 3.4.5. NIR-Laser Induced Photothermal Effect of AuNRs

Temperature increment (Δ*T*) tests were performed using a continuous wave fiber-coupled diode laser source of 808 nm wavelength (50 W, Oclaro, Inc., San Jose, CA, USA). The laser was powered by a Newport 5700-80 regulated laser diode driver (Newport Corporation, Irvine, CA, USA). A 200 μm-core optical fiber was used to transfer the laser light from the laser unit to the target solution and it additionally was equipped with a lens telescope mounting accessory at the output, which allowed the fine tuning of the laser spot size in the range of 0.1–10 mm. The laser spot size was measured with a laser beam profiler (Newport LBP-1-USB) placed at the same distance (8 cm) between the lens telescope output and the cuvette (or the 6-well plate), using the software Newport LBP series Measurement Systems v3.11 (Appendix A, as an example). In this way, the power per unit area was easily obtained. The final spot size was set at 1 cm in diameter. A potentiometer (Newport Optical Power Meter model 1916c) was used to calibrate the output power related to the intensity signal of the laser controller. The temperature of AuNR samples was measured with a type J thermocouple linked to a digital thermometer inserted into the solutions. Samples were irradiated for ca. 20 min, and then the laser was turned off and the temperature decay recorded. Particle solutions were stirred during laser illumination to homogenize the produced heat and ensure that samples were in thermal equilibrium during the entire course of the experiments. Water was used as control to determine the temperature increase (Δ*T*) due to laser exposure in the absence of the platform. The photothermal conversion efficiency (η) was determined by fitting experimental data to the model reported by Roper et al. [7]:(2)$$\Delta T\left(t\right)=\tau_{s}\left[\frac{I\left(1-10^{-A_{\lambda }}\right)\eta +Q_{0}}{\sum_{i}m_{i}c_{p_{i}}}\right]\left(1-e^{\frac{-t}{\tau_{s}}}\right)$$
where *T* is temperature, *t* is time, $m_{i}c_{p_{i}}$ is the specific heat capacity and mass of each component of the measuring system (platform dispersion, measuring cell, stirrer, etc.), $I\left(1-10^{-A_{\lambda }}\right)\eta$ is the heat dissipated by the electron–phonon relaxation process induced by light absorption and photothermal conversion, *I* is the laser power, *η* represents the efficiency of transducing light to thermal energy, $A_{\lambda }$ is the sample absorbance at maximum absorption wavelength, $Q_{0}$
is the heat produced by the laser on the measuring system determined with pure water, and $\tau_{s}$ is a characteristic rate constant defined as:(3)$$\tau_{s}=\frac{\sum m_{i}c_{p_{i}}}{hA}$$

On the other hand, the simulation of the photothermal effect generated by a single AuNR was made following Dyadic Green’s function to solve Maxwell’s equations for monochromatic fields when one AuNR with longitudinal LPSR band centered at 802 nm and dimensions 32 × 8 nm was excited under 808-NIR light excitation at 1.0 W/cm^2^. The AuNR NP and their water surroundings were discretized in a 150 × 150 elemental grid and subjected to a planar polarized (from 0 to 90°) illumination wave, and using the Lippmann–Schwinger equation for the electric field, *E* (*r*,*ω*):(4)$$E\left(r,\omega \right)=E_{0}\left(r,\omega \right)+\int G_{tot}^{EE}\left(r,r^{\prime },\omega \right)\chi_{e}E\left(r^{\prime },\omega \right)dr^{\prime }$$
where *E*_0_ (*r, ω*) is the incident electric field*,*
$G_{tot}^{EE}$ is Green’s dyad, and χ_e_ is the electrical susceptibility, one AuNR and their water surroundings were discretized in a 150 × 150 elemental grid and subjected to a planar polarized (from 0 to 90°) illumination wave, and the electric field at each element calculated. Using such values, the extinction, absorption and scattering cross sections could be calculated: $$\sigma_{ext}=\frac{4\pi k}{{\left|E_{0}\right|}^{2}}\overset{N_{cells}}{\underset{i=1}{\sum }}Im\left(E_{0,i}^{*}\cdot P_{i}\right)$$
(5)$$\sigma_{Abs}=\frac{4\pi k}{{\left|E_{0}\right|}^{2}}\overset{N_{cells}}{\underset{i=1}{\sum }}\left(Im\left(P_{i}\cdot E_{i}^{*}\right)-\frac{2}{3}k^{3}{\left|P_{i}\right|}^{2}\right)$$
$$\sigma_{scat}=\sigma_{ext}-\sigma_{abs}$$
where *k* is the wavenumber in the surrounding medium, and *E_i_* and *P_i_* are the electric field and polarization inside the nanostructure (at each discretized volume) induced by the applied electric field, *E*_0,*i*_, and in which the asterisk denotes complex notation. The energy release in the form of heat, *Q*(*ω*)*,* can be extracted by means of:(6)$$Q\left(\omega \right)=\frac{\omega }{8\pi }\overset{N_{cells}}{\underset{i=1}{\sum }}Im\left(\epsilon \left(r\right)\right){\left|E\left(r,\omega \right)\right|}^{2}V_{cell}$$
where *V_cell_* is the volume of the unit cell; and the temperature increment, Δ*T*, through the equation:(7)$$\Delta T\left(r_{probe},\omega \right)=\frac{1}{4\pi \kappa_{env}}\overset{N_{cell}}{\underset{i=1}{\sum }}\frac{q\left(r_{i},\omega \right)}{\left|r_{probe}-r\right|}V_{cell}$$
where κ_env_ is the thermal conductivity in the medium where the platform is located, and $q\left(r_{i},\omega \right)$ is the energy density in the form of heat generated by light excitation at each elemental volume.

#### 3.4.6. Colloidal Stability of AuNRs

The physical stability of the PE-coated AuNRs was assessed by diluting the samples (1/50) at 37 °C under moderate stirring at various pH ranging from 3.0 to 9.0 in the absence and presence of 10% (*v*/*v*) FBS, or trypsin. ζ-potential and DLS measurements were collected using a Zetasizer Nano ZS (Malvern, UK) for 7 days. The experiments were performed in triplicate for three different NP batches.

### 3.5. Nanoplatform Characterization

To determine the encapsulation efficiency (EE) of DOXO within PSS/PLL/DOXO/HA-coated AuNRs, the hybrid NPs were centrifuged at 15,000× *g* rpm at 20 °C for 20 min. The DOXO content in the supernatant was measured by means of UV-Vis and fluorescent spectroscopies. Previously, a calibration curve with free DOXO was obtained, and the absorbance/fluorescence of supernatants of bare AuNR solutions were also considered as an additional blank. UV-Vis measurements were made at 488 nm. Fluorescence standard curves were set at λ_ex_ = 480 nm and collected at λ_em_= 560–590 nm. UV-Vis spectra were measured in a Cary Bio 100 UV-Vis spectrophotometer (Agilent Technologies, USA). Fluorescence spectra were monitored in a Cary Eclipse spectrophotometer (Agilent Technologies, USA). Each sample was measured in triplicate for three different batches. The EE% was calculated by the following expression:(8)$$EE\left(\%\right)=\frac{Total\;amount\;of\;DOXO\;feeded-DOXO\;in\;supernatant}{Total\;amount\;of\;DOXO\;feeded}\times 100$$

### 3.6. In Vitro DOXO Release 

DOXO cumulative release profiles from the hybrid AuNRs/PSS/DOXO/PLL/HA NPs (1 mL, [c] = 1·10^11^ NP/mL) were measured in vitro at a constant temperature of 37 °C at 300 rpm magnetic stirring for several days at pH 7.4 and 5.0 in the absence and presence of trypsin–ethylenediaminetetraacetic acid (EDTA, 50 µL per 40 mL of buffer) and/or NIR light irradiation at 1.0 W/cm^2^ for 5 min at different time intervals using a 808 nm diode laser coupled to a 200 μm optical fiber connected to a collimator, which allowed us to select the size of the beam spot. To obtain the release profiles, 2 mL of hybrid AuNRs were placed inside dialysis tubes (SpectraPore^®^, MWCO 3500 Da) immersed into 50 mL of buffer supplemented with 10% (*v*/*v*) fetal bovine serum (FBS) at the pH of interest. The released DOXO concentration was determined at different time intervals. At each sampling time, ca. 0.5 mL of the medium was withdrawn and replaced with the same volume of fresh buffer to maintain the required sink conditions. The DOXO content was measured by means of UV-Vis and fluorescence using previously established calibration curves in the corresponding buffers. UV-Vis spectra were measured in a Cary Bio 100 UV-Vis spectrophotometer (Agilent Technologies, USA), and fluorescence spectra were monitored in a Cary Eclipse spectrophotometer (Agilent Technologies, USA). Assays were carried out in triplicate.

### 3.7. Tumor Cells

Tumoral cervical Hela and 3T3 Balb fibroblast cells were purchased from Cell Biolabs (San Diego, CA, USA) and grown at standard culture conditions (5% CO_2_ at 37 °C) in Dulbecco’s modified Eagles medium (DMEM) supplemented with 10% (*v*/*v*) FBS, 2 mM L-glutamine, 1% penicillin/streptomycin, 1 mM sodium pyruvate, and 0.1 mM non-essential amino acids (NEAA).

### 3.8. Cellular Uptake

Cellular uptake of the present hybrid AuNRs/PSS/PLL/HA AuNR nanoconstructs by HeLa and 3T3 Balb cells was investigated by TEM, ICP-MS and fluorescence microscopy. Cells were seeded in 6-well plates (2 mL, 5·10^4^ cells per well) and grown for 24 h at standard culture conditions. Then, 200 µL of PSS/PLL/HA-coated AuNR solution (2.5·10^10^ NP/mL) were added to cells. After 6 h of incubation, the NP-containing cells were washed three times with PBS, trypsinised, and centrifuged at 1200× *g* rpm for 4 min. Cell pellets were fixed with 500 µL of 2.5% wt. of glutaraldehyde. The pellets were then embedded in an agar pellet, postfixed with osmium tetraoxide in 0.1 M cacodylate buffer (1% (*w*/*v*)), and finally pelletised with Eponate (Ted Pella Inc, Redding, CA, USA). Ultra-thin cuts were obtained with an ultramicrotome (UltraCut S, Leica Microsystems GmbH, Germany) and were analyzed by TEM (JEOL JEM 1011, Japan). 

Next, cell uptake and internalization and DOXO release were additionally monitored by fluorescence microscopy. Cells were seeded on poly-L-lysine-coated glass coverslips (12×12 mm^2^) placed inside 6-well plates (2 mL, 5·10^4^ cells per well) and grown for 24 h at standard culture conditions. Then, 250 μL of hybrid NPs (2.5·10^10^ NP/mL) were added to cells. Some wells with cells were also irradiated with a continuous wave 808 nm fiber-coupled diode laser source (50 W, Oclaro Inc., San Jose, CA, USA) for 5 min at 0.5 W/cm^2^. After the desired incubation time, the NP-containing cells were washed three times with ice-cold PBS at pH 7.4, fixed with paraformaldehyde 4% (*w*/*v*) for 10 min, washed with PBS again, permeabilized with 0.2% (*w*/*v*) Triton X-100, washed again with PBS, mounted on glass slides, stained with ProLong Gold antifade DAPI (Invitrogen, Waltham, MA, USA), and cured for 24 h at −20 °C. Samples were visualized with a 63× objective using an epifluorescence microscope Leica DMI6000B (Leica Microsystems GmbH, Heidelberg Mannheim, Germany), where the blue channel corresponds to DAPI (λ_ex_= 355 nm), and the red channel to DOXO (λ_ex_ = 480 nm) and the transmitted light was observed in differential interference contrast (DIC) mode.

### 3.9. In Vitro Cell Citotoxicity

The cytotoxicity of AuNRs/PSS/DOXO/PLL/HA NPs was tested in vitro by means of the CCK-8 cytotoxicity assay. Cervical HeLa and 3T3 Balb fibroblast cells were seeded into 96-well plates (1·10^4^ cells/well) and grown for 24 h at an optical confluence of 80–90% under standard culture conditions in 100 μL growth medium. PSS/PLL/HA-coated and PSS/DOXO-coated AuNRs and free DOXO were used as negative and positive controls, respectively. After 24 h of incubation at 37 °C, 100 μL of NPs at 2.5·10^10^ NP/mL in the corresponding cell culture medium were injected into the wells and incubated for 24 h and 48 h. Some wells were tested in the absence of NPs as a negative control (blank), and free DOXO at the same concentration as that encapsulated in the hybrid AuNR nanoplatform. After the corresponding incubation, the culture medium was discarded, cells were washed with 10 mM PBS (pH 7.4) several times, and fresh culture medium (100 μL) containing 10 μL of CCK-8 reagent was added to each well. After 2 h, the absorption at 450 nm of cell samples was measured with a UV-vis microplate absorbance reader (Bio-Rad model 689, Hercules, CA, USA). Cell viability (*SR*, survival rate) was calculated as follows:(9)$$SR=\frac{Abs_{sample}}{Abs_{blank}}\times 100$$
where *Abs_sample_* is the absorbance at 450 nm for cell samples and *Abs_blank_* is the absorbance corresponding to the sample controls without the particles. In addition, some of the wells were also irradiated with a continuous wave fiber-coupled diode laser source at 808 nm (50 W, Oclaro, Inc., San Jose, CA, USA). The used power fluencies were 0.5, 1.0, and 3.0 W/cm^2^ for 5 min. After 18 h and 42 h, cells were washed again, and new fresh culture medium (100 μL) was added with 10 μL of CCK-8 reagent to each well and measured as specified above. Experiments were repeated at least three times.

### 3.10. Cell Death Mechanism

Cervical HeLa and 3T3 Balb fibroblast cells were seeded into 96-well plates (1·10^4^ cells/well) and grown for 24 h at an optical confluence of 80–90% under standard culture conditions in 100 μL growth medium. Next, 100 μL of AuNRs/PSS/PLL/HA and AuNRs/PSS/DOXO/PLL/HA encapsulating 75 and 100 μg DOXO were added and incubated for 6 h. 2 μg/mL of free DOXO and non-treated cells were used as positive and negative control for apoptosis, respectively. After incubation, the culture medium was changed by fresh one and cells were illuminated with an 808 nm CW laser at 1.0 and 2.0 W/cm^2^ for 5 min. After 48 h, cells were trypsinized, redispersed in 500 μL of fresh medium (7.5·10^4^ cells/mL) and centrifuged at 1200× *g* rpm for 4 min. Then, 100 μL of cells were mixed with 200 μL of buffer solution and 10 μL of Annexin V and 10 μL of Propidium iodide reagents, followed by incubation for 15 min in the dark at room temperature. Finally, cells were analyzed by flow cytometry (Luminex Guava easyCyte 12HT, Luminex Corp, Austin, TX, USA).

### 3.11. Statistical Analysis

Statistical analysis of experimental data was performed with Origin software v.8.0. All results were presented as mean standard deviation unless otherwise stated. One-way ANOVA (* *p* < 0.05; ** *p* < 0.01, *** *p* < 0.005) was used to determine statistical differences for multiple groups, whereas unpaired *t*-test was used to analyzed individual groups.

## 4. Conclusions

In summary, AuNRs in high yields with tunable NIR LPSR maxima from 700 to 816 nm and particle lengths and widths within a very narrow range of 32–36 and 8–12 nm (AR variation from ca. 2.6 to 4.1), respectively, were obtained. Surprisingly, the loss of heat generation efficiency by 70 nm blue shifted off-resonant LPSRs regarding the CW excitation laser source (808 nm) used, thus providing suitable photothermal responses at both low and moderate particle concentrations and light fluencies. To provide cytocompatibility as well as colloidal stability, Au NRs of 802 nm were surface-functionalized through the sequential adsorption of PSS, PLL and HA polymers using the LbL assembly technique, the latter one providing targeting capability to CD44 receptors overexpressed in different tumoral cells. Prior to PLL deposition, the chemo-drug DOXO was incorporated in the polymeric coating up to a concentration of ca. 350 μg/mL through an optimized process. DOXO release was observed to be negligible at the early stages of incubation thanks to the compact packing of the polymeric surface coating and simultaneously triggered by the enzymatic degradation of the PLL layer by physiological proteases and the external applied NIR light, giving rise to stair-like release profiles which enable a perfect control over the cargo diffusion out of the nanoplatform. Finally, the analysis of the multimodal therapeutic ability of the developed nanoplatform demonstrated that the combination of both chemo- and photothermal therapy provide an outstanding targetable cytotoxicity to cancerous malignant cells, thanks to their efficient uptake promoted by recognition of CD44 receptors overexpressed on their membrane surfaces. Conversely, such toxic effect is not observed in healthy cells used as control. Moreover, it was observed that the combined toxic effect provided by photothermal and chemotherapy is mainly driven by apoptosis, which might suggest that cell death is initiated/dominated by the toxic effect of the drug supported by the localized photothermia, as observed from the experiments done at different solution temperatures. This also agrees with the necrotic pathway induced when only light is used to eradicate the tumoral cells.

## Figures and Tables

**Figure 1 ijms-23-13109-f001:**
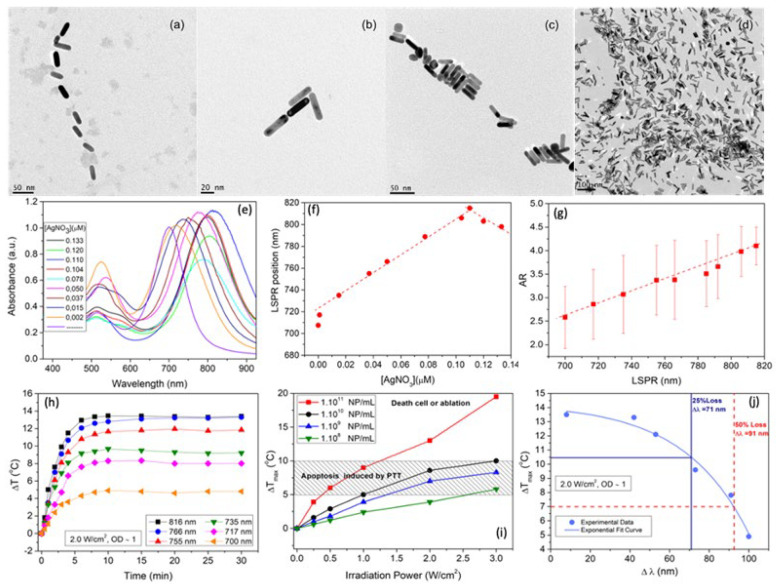
TEM images of synthesized AuNRs with dimensions: (**a**) 34 × 12, (**b**) 36 × 11, (**c**) 34 × 10, (**d**) 31 × 8 nm, obtained after the addition of 0.012, 0.043, 0.032 and 0.096 µM of AgNO_3_ to the Au-CTAB complex, respectively. Changes in (**e**) absorption spectrum; (**f**) LSPR maximum; and (**g**) AR of AuNRs upon changes in AgNO_3_ concentration during colloidal synthesis. Dashed lines are only to guide the eye. In (**e**), the two classical extinction peaks are observed: the transverse LSPR shows little changes with AuNR AR (ca. 510 nm), whilst the longitudinal one markedly redshifts as the particle AR increases (from 700 to 816 nm). (**h**) Temperature increments (Δ*T*) for AuNRs with LSPRs located at (■) 816 nm, [33.1 × 8.1 nm], (●) 766 nm [36.6 × 11.1 nm], (▲) 755 nm [34.0 × 10.2 nm], (▼) 735 nm [30.2 × 10.1 nm], (◆) 717 nm [33.5 × 11.9 nm] and (◀) 700 nm [34.4 × 12.4 nm] using a CW laser source of 808 nm at 2.0 W/cm^2^. (**i**) Solution temperature dependence of CTAB-coated AuNRs with a LSPR band centered at 802 nm at different particle concentrations under NIR illumination (808 nm, 30 min) of different intensities. (**j**) AuNRs efficiency loss as a function of the LSPR-resonance shift. This shift was calculated as Δλ = │λ_laser_ − λ_SPRL_│.

**Figure 2 ijms-23-13109-f002:**
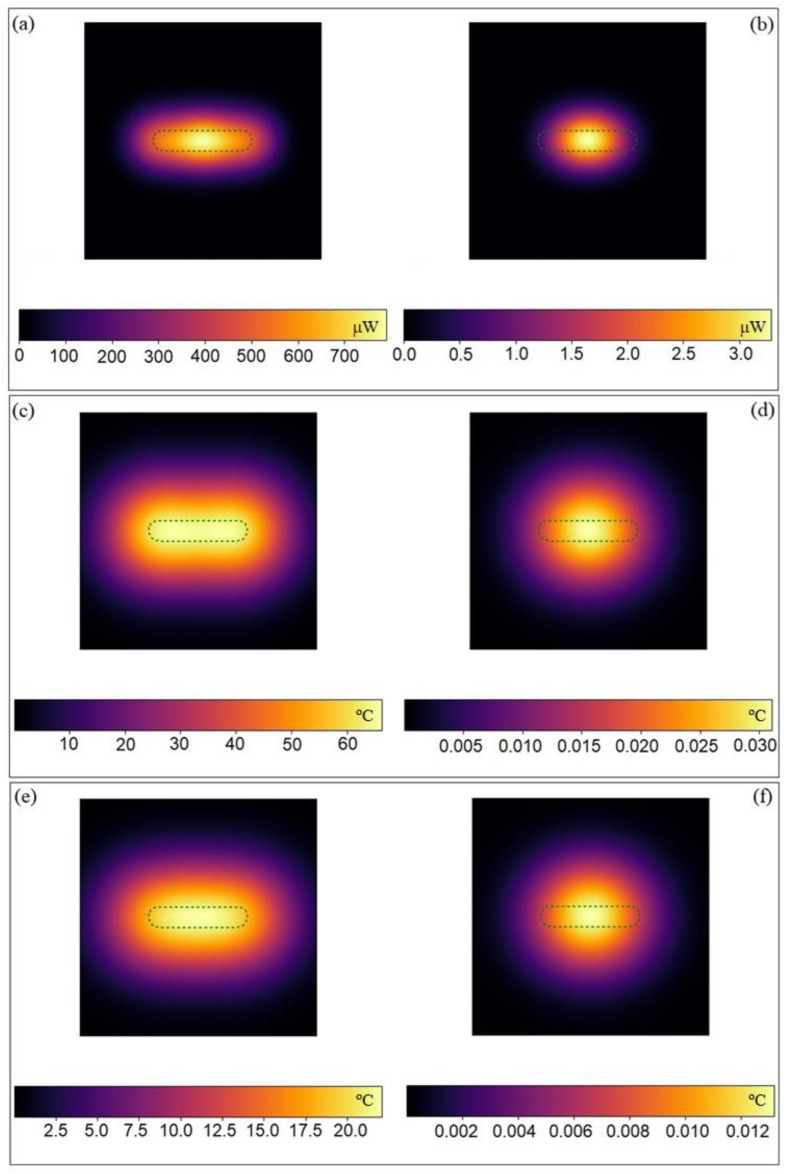
Released energy map from an AuNR illuminated with a laser light of 808 nm at 1 W/cm^2^ for 5 min with (**a**) 0° and (**b**) 90° polarization. Heating maps of an AuNR illuminated with a laser light of 808 nm at 1 W/cm^2^ for 5 min with 0° (**c**,**e**) and 90° (**d**,**f**) polarization at a distance from the observer of 5 (**c**,**d**) and 25 (**e**,**f**) nm along the perpendicular plane.

**Figure 3 ijms-23-13109-f003:**
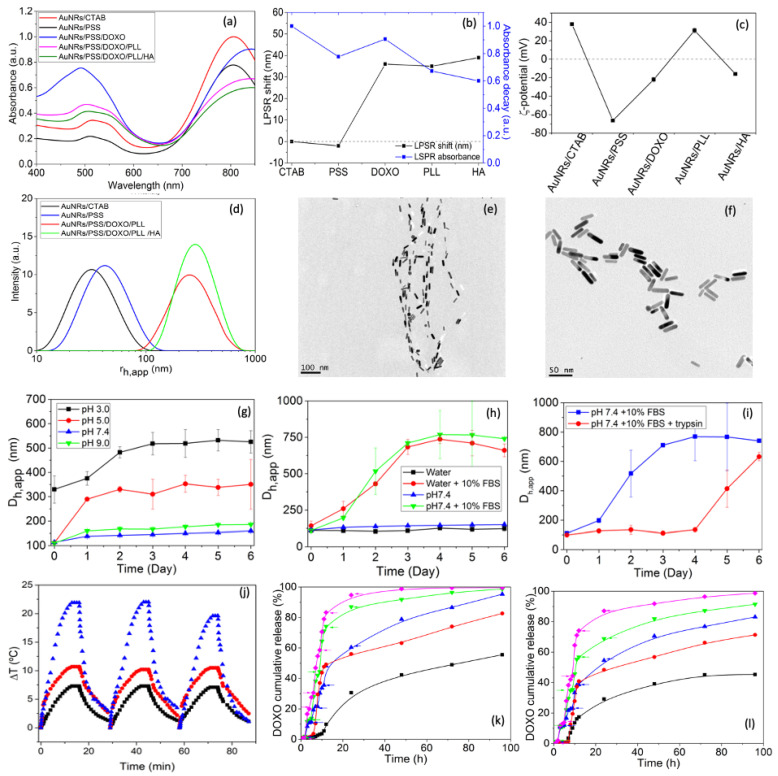
Changes in (**a**) absorption spectra, (**b**) LPSR absorbance and position, (**c**) surface charge, and (**d**) population size distributions of AuNRs after the different polymeric coating steps. TEM images of (**e**) CTAB-coated and (**f**) PSS/PLL-DOXO/HA-coated AuNRs. Colloidal stability of PSS/PLL/HA-coated AuNRs by monitoring the time evolution of platform hydrodynamic sizes at (**g**) different pH, (**h**) in the absence and presence of serum, and (**i**) of trypsin at 37 °C. Samples were diluted in buffers of pH 3.0 (■), 5.0 (●), 7.4 (▲), and 9.0 (▼). (**j**) Temperature increments after successive irradiation cycles at (■), 1.0 (●), 2.0 (▲) and 3.0 W/cm^2^ using an 808 NIR laser light. DOXO release profiles from PSS/DOXO/PLL/HA-coated AuNRs at (**k**) pH 5.0 and (**l**) 7.4 alone (■), or in the presence of trypsin (●), NIR light (1.0 W/cm^2^ for 5 min, at 2, 5, 9, 12 and 24 h (▲), trypsin + NIR light of 1.0 W/cm^2^ (similar time points, ▼) and trypsin + NIR light of 2.0 W/cm^2^ (similar time points, ◆). Errors bars in release plots are not shown for clarity, but uncertainties are below 15%. Arrows denote the time points of NIR light application.

**Figure 4 ijms-23-13109-f004:**
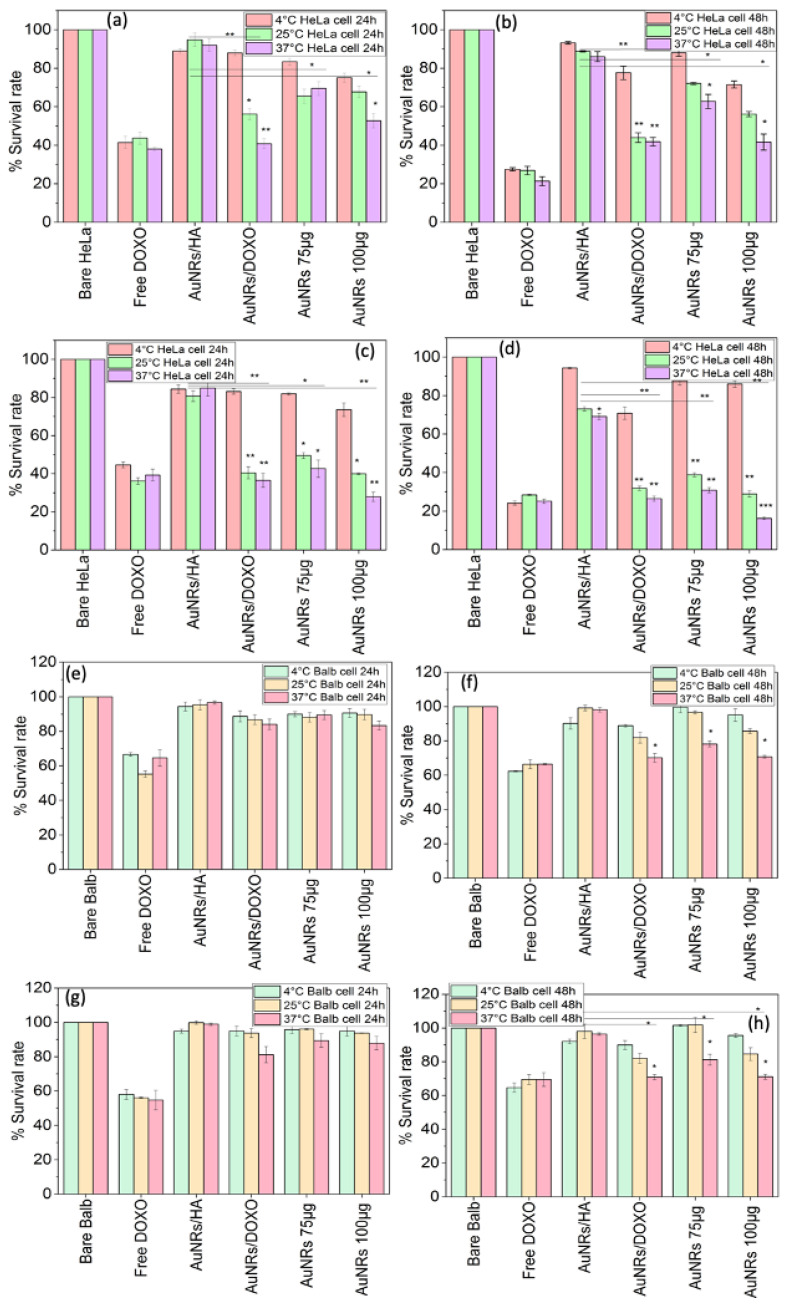
Cellular cytotoxicity expressed as survival rate of (**a**–**d**) HeLa and (**e**–**h**) 3T3 Balb cells at different incubation times and temperatures in the presence of NIR light illumination of (**a**,**b**,**e**,**f**) 1.0 and (**b**,**d**,**f**,**h**) 2.0 W/cm^2^. AuNRs/HA denote PSS/PLL/HA-coated NPs, AuNRs/DOXO correspond to PSS/DOXO-coated AuNRs, and AuNRs 75 µg and AuNRs 100 μg stand for PSS-/DOXO-PLL/HA-coated AuNRs encapsulating 75 and 100 µg of the drug, respectively. Statistical significance compared to 4 °C at each group: * = *p* < 0.05; ** = *p* < 0.01; *** = *p* < 0.005.

**Figure 5 ijms-23-13109-f005:**
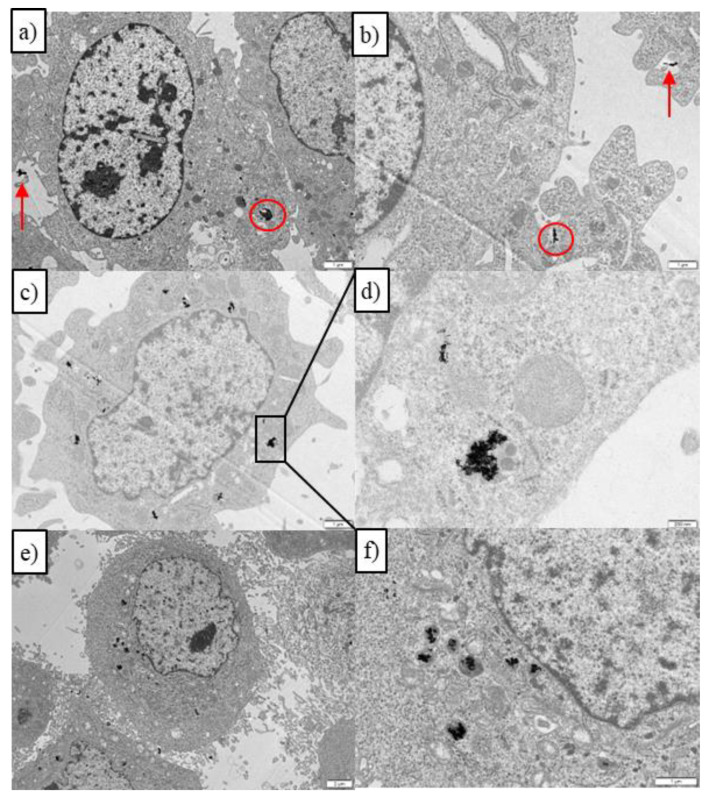
TEM images of PSS/DOXO/PLL/HA-coated AuNRs inside (**a**,**b**) 3T3 Balb in the absence of NIR illumination, and HeLa cells (**c**,**d**) in the absence and (**e**,**f**) presence of NIR illumination (808 nm, 1.0 W/cm^2^ for 5 min). (**d**) depicts a zoomed endocytic vesicle containing numerous AuNRs. In (**a**,**b**), red circles denote particles inside cells endosomes. Red arrows denote particles in the surroundings of the cell membrane. Scale bars are 1 μm.

**Figure 6 ijms-23-13109-f006:**
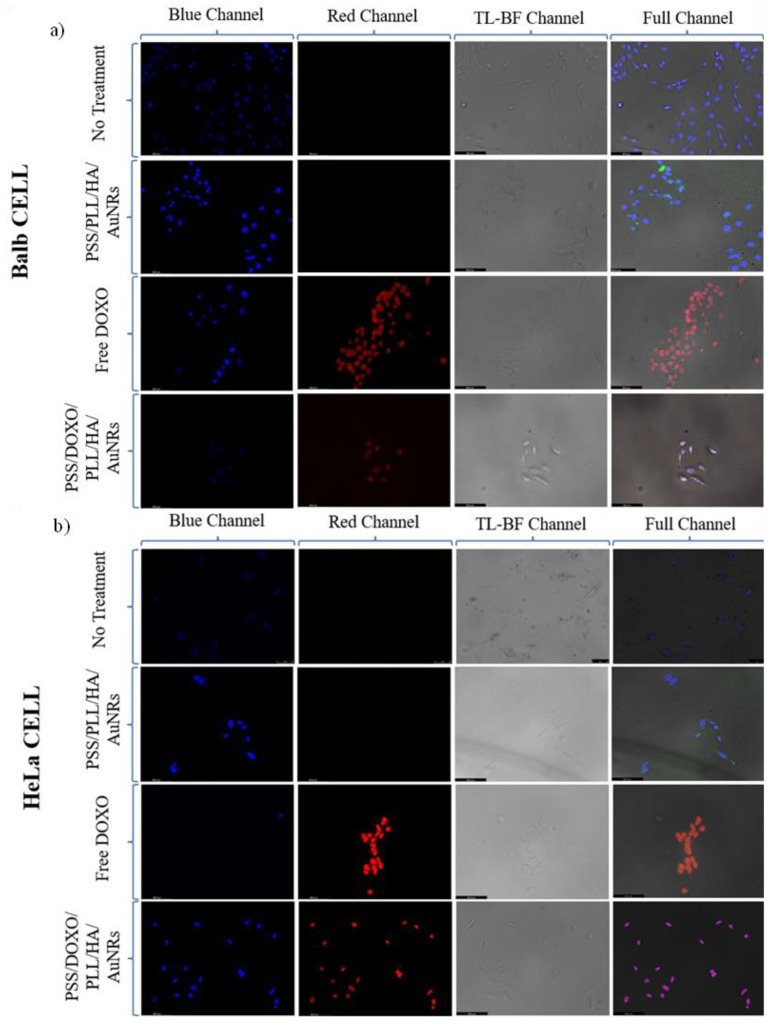
Fluorescence microscopy images of PSS/DOXO/PLL/HA-coated AuNRs inside (**a**) 3T3 Balb and (**b**) HeLa cells at 12 h of incubation. Images shown blue, red and bright field channels, as well as the merged images: Blue channel displays cell nuclei stained with DAPI (λ_ex_ = 355 nm); the red one DOXO fluorescence (λ_ex_ = 488 nm), and the bright field the optical image. Scale bars are 10 µm.

**Figure 7 ijms-23-13109-f007:**
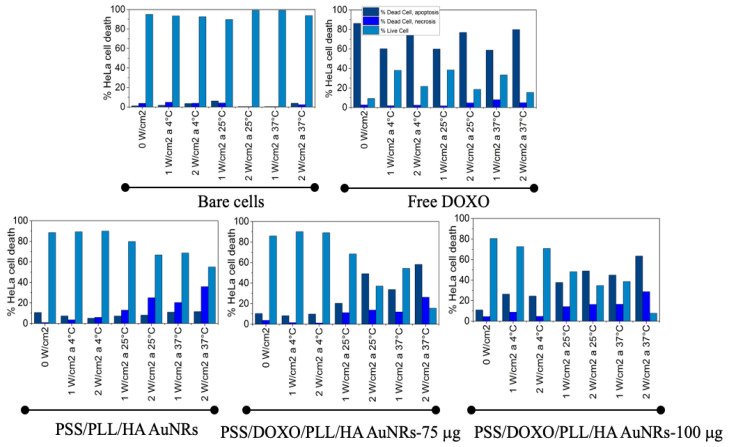
Main cell death mechanisms involved in cell mortality after administration of free DOXO, bare PSS/PLL/HA-coated and PSS/DOXO/PLL/HA-coated AuNRs with 75 and 100 µg DOXO for HeLa cells and at different temperatures and NIR fluencies.

**Table 1 ijms-23-13109-t001:** Photothermal conversion efficiency and relaxation time of PSS/PLL/HA-coated AuNRs after the different irradiation cycles at different fluencies.

Fluency (W/cm^2^)		η	τ_s_ (s)
1.0	Cycle 1	0.45 ± 0.01	9.01 ± 0.58
	Cycle 2	0.46 ± 0.02	8.95 ± 0.55
	Cycle 3	0.44 ± 0.02	9.23 ± 0.62
2.0	Cycle 1	0.67 ± 0.04	3.90 ± 0.28
	Cycle 2	0.65 ± 0.03	4.05 ± 0.29
	Cycle 3	0.68 ± 0.09	3.80 ± 0.28
3.0	Cycle 1	0.55 ± 0.03	7.84 ± 0.61
	Cycle 2	0.56 ± 0.04	8.01 ± 0.91
	Cycle 3	0.49 ± 0.03	7.19 ± 0.80

## Data Availability

Not applicable.

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
