# Peer review of "Hybrid Gold Nanorod-Based Nanoplatform with Chemo and Photothermal Activities for Bimodal Cancer Therapy"

_ijms, 2022, doi:10.3390/ijms232113109_

Round 1

Reviewer 1 Report

Comments regarding the manuscript ijms-1953660, entitled “Hybrid gold nanorod-based nanoplatform with chemo and photothermal activities for bimodal cancer therapy” by Arellano et al.

The work is well-structured, the experimental results are discussed in detail and the conclusions appropriate. In my opinion, the paper is suitable for publication in International Journal of Molecular Sciences after minor revisions:

Figures 1, 3, 4, 7, S5, S7, S10 need to be significantly improved as axes and legends font is too small and reading results very difficult.

Author Response

Reviewer 1

We thank the reviewer by his/her comments which help to improve the quality of the manuscript.

  1. Figures 1, 3, 4, 7, S5, S7, S10 need to be significantly improved as axes and legends font is too small and reading results very difficult.

Answer (A): We thank the reviewer by his/her observation. Figures have been revised and improved according to the reviewers´ comments. In addition, Figure 7 has been split and a new Figure S13 included in Supporting Information to gain clarity in the main text.

Reviewer 2 Report

1) The characterization of materials is not complete, which need improve by adding the dynamic light scattering, zeta potential and TEM results. Need demonstrate the various components of the nanorpobes were really integrated as a water soluble drugs.

2) The grammar errors must be corrected. For example, for the first sentence in the introduction, "etc" should be ", and etc." And the second, and third sentences also have grammar errors.

3) The cell viability with and without laser illumination for normal and cancers should be carried out for the probe.

4) Better offer  in vivo imaging and cancer treatment results. 

5) Histology analysis better be provided for in vitro study. 

Author Response

We thank the reviewer by his/her comments which help to improve the quality of the manuscript.

  1. The characterization of materials is not complete, which need improve by adding the dynamic light scattering, zeta potential and TEM results. Need demonstrate the various components of the nanorpobes were really integrated as a water soluble drugs.

Answer: We thank the reviewer by his/her comment. However, data requested by the reviewer were already present in the manuscript: DLS, Figure 3d for size, and Figure g,h,I to analyze stability; TEM, Figure 1 a-d and Figure 2 e,f; and zeta potential, Figure 3c and Figure S7. In this regard, precisely this manuscript is based on the development and full characterization of the nanosystem, and an initial biological evaluation in vitro to decipher the role played by chemo and phototherapy in cytotoxicity and cell death.

  1. The grammar errors must be corrected. For example, for the first sentence in the introduction, "etc" should be ", and etc." And the second, and third sentences also have grammar errors.

Answer: We thank the reviewer for the observation. Several grammar and typo mistakes have been corrected along the text.

  1. The cell viability with and without laser illumination for normal and cancers should be carried out for the probe.

Answer: We thank the reviewer by his/her comment. However, data requested by the reviewer were already present in the manuscript, see Figure 4 and Figures S9 and S10.

  1. Better offer in vivo imaging and cancer treatment results.

 Answer: We thank the reviewer for the observation. However and as commented in Answer 1, the present manuscript is focused on the development and full characterization of the nanosystem, and an initial biological evaluation in vitro to decipher the role played by chemo and phototherapy in cytotoxicity and cell death. A more extensive in vitro study under dynamic conditions (microfluidics mimicking models) and 3D cell cultures as well as in vivo using suitable anima tumoral model is being performed to assess the therapeutic efficacy of the proposed combined phototherapy, which will be the subjected of a forthcoming publication.

  1. Histology analysis better be provided for in vitro study.

 Answer: Reviewer here should be wrong, since in vitro analysis was made in cell cultures and not with tissues, so histology cannot be possible to be done. Instead, a deep confocal microscopy analysis was performed.

Round 2

Reviewer 2 Report

Can be accepted now.